# Extracellular Inorganic Phosphate-Induced Release of Reactive Oxygen Species: Roles in Physiological Processes and Disease Development

**DOI:** 10.3390/ijms22157768

**Published:** 2021-07-21

**Authors:** Marco Antonio Lacerda-Abreu, José Roberto Meyer-Fernandes

**Affiliations:** 1Instituto de Bioquímica Médica Leopoldo de Meis, Universidade Federal do Rio de Janeiro, Rio de Janeiro 21941-901, RJ, Brazil; 2Instituto Nacional de Ciência e Tecnologia em Biologia Estrutural e Bioimagem, Universidade Federal do Rio de Janeiro, Rio de Janeiro 21941-590, RJ, Brazil

**Keywords:** inorganic phosphate, hyperphosphataemia, reactive oxygen species

## Abstract

Inorganic phosphate (Pi) is an essential nutrient for living organisms and is maintained in equilibrium in the range of 0.8–1.4 mM Pi. Pi is a source of organic constituents for DNA, RNA, and phospholipids and is essential for ATP formation mainly through energy metabolism or cellular signalling modulators. In mitochondria isolated from the brain, liver, and heart, Pi has been shown to induce mitochondrial reactive oxygen species (ROS) release. Therefore, the purpose of this review article was to gather relevant experimental records of the production of Pi-induced reactive species, mainly ROS, to examine their essential roles in physiological processes, such as the development of bone and cartilage and the development of diseases, such as cardiovascular disease, diabetes, muscle atrophy, and male reproductive system impairment. Interestingly, in the presence of different antioxidants or inhibitors of cytoplasmic and mitochondrial Pi transporters, Pi-induced ROS production can be reversed and may be a possible pharmacological target.

## 1. Introduction

Inorganic phosphate (Pi) is a simple chemical element essential for ATP formation through glycolysis, gluconeogenesis, or energy metabolism or as a constituent of DNA, RNA, phospholipids, and a variety of phosphorylated metabolic intermediates [1,2,3,4]. Healthy adults have a recommended dietary allowance of at least 700 mg of phosphorus per day, varying according to the food group ingested per day (Table 1) [5]. Approximately 85% of total body phosphorus is found in bone, where it is mainly complexed with hydrogen, oxygen, and calcium (Ca^2+^) to form apatite crystals deposited in the collagen matrix. The remainder of phosphorus is found in soft tissue (14%), with only approximately 1% in extracellular fluids [5,6,7].

Pi absorption is possible due to transporter-mediated translocation across cell membranes. Pi enters cells via Na^+^/Pi and H^+^/Pi cotransporters [1,2,3]. Na+/Pi cotransporters are comprised of two large families in mammalian cells, SLC20 and SLC34 [2,3,4,8]. The SLC20 family is composed of two members, i.e., PiT-1 (encoded by SLC20A1) and PiT-2 (encoded by SLC20A2), which can act as extracellular Pi sensors and are widely expressed in kidney, brain, lung, heart, liver, muscle, and osteoblast cells [1,2,5]. The SLC34 family contains three members, namely, NaPi-IIa (SLC34A1) expressed in the kidney and lung, NaPi-IIb (SLC34A2) expressed in the small intestine and lung, and NaPi-IIc (SLC34A3) expressed in the kidney, which is strictly localized in the apical brush-border membrane of renal proximal tubular cells [1,2].

Although Na^+^-dependent Pi transporters have been widely explored, another class of sodium-independent Pi transporters plays a significant role in Pi homeostasis and has been identified and characterized in several tissues associated with pathological conditions [3]. Inorganic phosphate is maintained in equilibrium in the range of 0.8–1.4 mM Pi, which is mediated by sodium-dependent Pi transporters, sodium-independent Pi transporters, and endocrine factors such as parathyroid hormone (PTH) and metabolically active vitamin D hormone (1α,25[OH]2D), fibroblast growth factor-23 (FGF-23) and a criticalcoreceptor α-klotho for FGF-23 signalling [7,9,10]. FGF-23 is produced in bone by osteocytes, whereas the parathyroid glands produce PTH. Both hormones activate their specific receptors on the proximal renal tubules to reduce the apical expression of the cotransporters NaPi-IIa and NaPi-IIc. PTH further increases the expression of the renal 1-α hydroxylase necessary for vitamin D production, while FGF-23 inhibits it. Thus, FGF23 and PTH have opposite effects on 1α,25[OH]2D production in the kidney, affecting the Pi released in urine. In addition, 1α,25[OH]2D production by kidneys enhances NaPi-IIb expression and Pi uptake in the gut [9,11].

Hyperphosphataemia is one of the main causes of morbidity and mortality in patients with chronic kidney disease (CKD) [12]. Hyperphosphataemia is caused by phosphate overloading and results in increased FGF-23 activity, decreased 1α,25[OH]2D production, and increased renal phosphate excretion [9,13]. In the renal tubule, a low expression of transmembrane-α-klotho is generally associated with kidney tubular cell resistance to FGF23, leading to hyperphosphatemia [10]. Hyperphosphataemia can be classified according to the level of serum Pi in the patient: mild (1.44–1.76 mM), moderate (1.76–2.08 mM), or severe (>2.08 mM) [11]. Several studies have reported hyperphosphataemia in association with hormonal changes or the worsening of other diseases, mainly due to the interference of Pi overload in cell signalling or even in the induction of oxidative stress [10].

Oxidative stress is classically defined as an event resulting from the magnitude of imbalance between oxidant and antioxidant substances generated in the context of oxidation-reduction reactions [14,15]. Commonly known as free radicals, oxidants include reactive oxygen and nitrogen species, such as superoxide (O_2_^−^), hydrogen peroxide (H_2_O_2_), nitric oxide (NO), and peroxynitrite (ONOO^–^) [14,15]. Regarding reactive oxygen species (ROS), H_2_O_2_ and O_2_^−^, are key redox signalling agents generated mainly by NADPH oxidases (NOX) and the mitochondrial electron transport chain (ETC) [13,14,15]. The mitochondrial electron transport chain leaks electrons from complexes I, II, and III to molecular oxygen to generate superoxide, which is rapidly converted into H_2_O_2_ [16].

In some human tissues (brain, liver, and heart), Pi has been shown to induce mitochondrial ROS release. Since Pi is the main anion permeable to intracellular membranes, it can alter mitochondrial pH gradients (∆pH) through the mitochondrial H^+^/Pi cotransporter and increase the electric membrane potential (∆Ψm), thus promoting ROS generation [17].

In this work, we highlight studies regarding the ability of Pi to induce the production of ROS and thus regulate several pathophysiological processes that result from different sources producing ROS, which are generally reversed by the addition of different antioxidants or when Pi uptake is inhibited in the cytoplasm and mitochondria.

## 2. Pi-Induced ROS Production Promotes Osteoblast Apoptosis

Serum phosphate concentration varies with age, with the highest concentration found in infants (1.50–2.65 mM Pi), who require more of the mineral for bone growth and soft tissue development. In adulthood, Pi concentrations are low (0.8–1.5 mM Pi), considering that most phosphate is present in bone and teeth compared to other tissues [18,19].

Bone growth can be regulated by the action of osteoblasts, and the maintenance of adequate Pi levels is crucial to the activity of osteoblasts in the process of matrix mineralization [3,19]. Therefore, the possibility exists that in the localized remodelling area, osteoblasts are exposed to levels of Pi that are substantially greater than those typically found in serum. In addition, osteoblasts are characterized by high alkaline phosphatase (ALP) activity, and this enzyme can elevate the local Pi concentration (Figure 1) [19,20]. Considerable numbers of studies have investigated whether an elevation in Pi promotes bone cell differentiation or apoptosis, mainly by Pi-induced ROS production in osteoblasts cells [19,20,21,22,23,24] (Table 2).

Although Pi is a necessary component of hydroxyapatite, Pi has been shown to participate in osteoblast differentiation signalling, which is marked by increased ALP expression, matrix vesicle formation, and hydroxyapatite mineral deposition [7]. When high Pi levels are achieved in bone mineralization, a decrease in ALP activity in osteoblasts is observed (Table 1) [20,21].

In osteoblastic murine MC3T3-E1 cells, high Pi levels suppress osteoblastic differentiation by Pi-induced ROS production (Table 2) [20]. The authors of this study point to NOX as the primary source of ROS-induced Pi after tests with NOX inhibitors (diphenyliodonium [DPI] and apocynin), and the use of silenced MC3T3-E1 cells for NOX isoforms (Nox1 and Nox4) suppresses phosphate-induced ROS production. In addition, PFA abrogates Pi-induced ROS production, suggesting that Pi enhances ROS production by entering cells [20]. The possibility of xanthine oxidase and the mitochondrial respiratory chain participating in Pi-induced ROS production was ruled out after no effect was observed in the presence of their respective inhibitors (10 μM oxypurinol or 10 μM rotenone) [20].

In a rat osteoblastic cell line (UMR-106), high Pi levels also induce ROS production through NOX, as shown by the sensitivity of this effect to apocynin (Table 2). However, Pi-induced ROS production seems to be involved in hormonal FGF-23 release [21]. In hyperphosphataemia, it is classically known that FGF-23 release is mainly coordinated by bone tissue in the blood. FGF-23 acts to inhibit the internalization of intestinal Pi and renal reabsorption and stimulates the excretion of Pi in the urine, promoting the reduction in serum Pi levels. In vitro experiments using UMR-106 cells have shown that phosphate directly regulates FGF-23 without affecting the stability of FGF-23 messenger RNA by stimulating NOX-induced ROS production and the MEK-ERK pathway [21].

In human bone cells, it has been demonstrated that Pi (7 mM Pi) serves to regulate osteoblast apoptosis and possibly depends on cytosolic Pi, since treatment with the Pi transporter inhibitor PFA reverses the effect of Pi-induced cell death [22]. Furthermore, it has been suggested that Pi treatment induces a mitochondrial membrane permeability transition (Table 2 and Figure 1) [22].

In bone, phosphate is primarily complexed with calcium in hydroxyapatite crystals; the remaining phosphate appears as amorphous calcium phosphate [18,19]. Since Ca^2+^ and Pi are released from the bone apatite lattice during the resorption process, Ca^2+^ may influence Pi-mediated bone cell apoptosis.

Therefore, the role of calcium and phosphate in the regulation of cellular apoptosis in osteoblasts (primary human osteoblasts and MC-3T3-E1 cells) has been investigated. In the presence of the ion pair (5 mM Pi and 2.9 mM Ca^2+^ treatment), the cells exhibit the characteristics of apoptosis, including contraction of the cells, the release of osteoblasts from the substratum, loss of contact with neighbouring cells and a marked change in the mitochondrial membrane potential (Table 2) [23]. The results indicated that when the calcium concentration is raised from 1.8 to 2.9 mM, there is a dramatic increase in Pi-mediated osteoblast death. Additionally, inhibitors of Ca^2+^ channel transport exert little effect on osteoblast apoptosis mediated by the ion pair, with some protection provided but only at high concentrations (100 μM) by a general Ca^2+^ channel inhibitor (lanthanum chloride) [23].

Mitochondrial dysfunction results in the release of Ca^2+^ from mitochondria. Consequently, an increase in intracellular Ca^2+^ accompanied by the loss of cytochrome c serves to activate events downstream in the apoptotic process [23]. In addition, there are other mechanisms by which Ca^2+^ and Pi in mitochondria mediate cell death. Initial hyperpolarization of the mitochondrial signal is observed in osteoblastic cells after treatment with Pi and Ca^2+^ for 2 h (Table 2 and Figure 1) [23]. However, 4 h after treatment, the mitochondrial membrane potential drops, and an increase in intracellular Ca^2+^ fluorescence was noted (Table 2 and Figure 1) [23].

Because the generation of ROS has also been linked to the activation of apoptotic pathways, the same research group showed that treatment with Pi (8 mM) and Ca^2+^ (2.9 mM) promotes an increase in the production of ROS for up to 45 min of treatment (Table 2), after which there was a return to baseline levels [24].

## 3. Phosphate-Induced Chondrocyte Apoptosis Is Mediated by Ros Production

Most bones form through endochondral ossification. This process involves a cartilage intermediate or template (cartilage growth plate or epiphyseal plate) on which the final bone forms. Within this process, chondrocytes undergo a well-organized series of maturation steps of growth plate physiology, ultimately resulting in hypertrophy [25]. In the growth plate, the cells terminally differentiated before zone calcification undergo apoptosis or programmed cell death (Figure 2). At cartilage mineralization sites, Pi levels are known to increase to approximately 2.20 mM Pi [26,27]. The amount of Pi used to evoke cell death at the location where chondrocytes initiate the apoptosis process is not known. However, it would not be unreasonable to assume that just before the deposition of the mineral, the local concentration of Pi would be high [28,29]. This Pi level would be sufficient to induce the transitional state in growth plate mitochondria and induce apoptosis in hypertrophic chondrocytes, as demonstrated in some studies usually using chick embryo chondrocytes, an experimental model system obtained by removing proximal tibial growth plates (Figure 2) (Table 3) [28,29,30,31,32].

A positive label for mitochondrial membrane potential is shown in chondrocytes in the proliferative zone compared to the hypertrophic zone. In the hypertrophic zone, the fluorescence yield is only marginally affected by Pi treatment for 24 h. Therefore, these results suggest that high Pi triggers apoptosis in these energy-compromised cells by promoting a mitochondrial membrane transition, thereby inducing the death process (Table 3) [28].

In embryonic tibial chondrocytes, it has been demonstrated that a high Pi concentration can induce dead cells at a late maturational stage that does not occur in cephalic and caudal chondrocytes (Table 3) [29]. Analysis of the death process has indicated apoptosis induced by high Pi. This effect may be linked to chondrocyte death, matrix vesicle biogenesis, and cartilage mineralization [29].

Based on the observation that Na-Pi transporters are expressed in chondrocytes, another study sought to identify whether inhibition of Na-Pi cotransport function can prevent Pi-induced apoptosis. The effect of PFA in conjunction with bisphosphonate and alendronate, both competitive inhibitors of Na-Pi cotransport, show a blockade of the effect induced by Pi in the cell apoptosis and Pi transport assays [30].

The co-relationship of Pi with Ca^2+^ in inducing apoptosis in chondrocytes has also been demonstrated after observing that EDTA and EGTA blocks Pi-induced apoptosis and Pi uptake in chondrocytes in a dose-dependent manner [31]. While high levels of Ca^2+^ alone have little effect, the cation enhances Pi-dependent cell death and significantly increases Pi uptake (Table 3) [31]. One explanation for the observation that Ca^2+^ synergizes with Pi-dependent apoptosis is that Ca^2+^ positively modulates Pi transporters in chondrocytes (Figure 2). An alternative explanation is that Ca^2+^ activates apoptosis by generating calcium phosphate nuclei, interacting with membrane receptors, and activating the death receptor complex (Figure 2). In that case, pro-caspase-8 would be converted to caspase-8, which causes a transition of the mitochondrial membrane and the processing of caspase-3 (Figure 2) [31].

In addition, to induce changes in mitochondrial function, the ion pair stimulates ROS generation, possibly followed by the mitochondria becoming hyperpolarized after 75 min by adding Pi and Ca^2+^ (Figure 2). This finding lends credence to the hypothesis that ROS generation combined with the loss of protective thiols is an early event in the apoptotic process (Table 3) [31].

NO stimulates apoptosis in several cell types, including chondrocytes [33]. In terms of the apoptotic cascade, NO can directly activate downstream effectors or catalyse the formation of other highly reactive intermediates, such as ONOO_2_^−^, that induce caspase 3 and other caspases [34]. Therefore, incubation with high Pi for 24 h induces apoptosis mediated by a decrease in ∆Ψm and ONOO_2_^−^ production (Table 3). These conclusions are supported by the observation that when cells are pretreated with an NO synthase (NOS) inhibitor (NG-nitro-L-arginine methyl ester: L-NAME or L-NMMA) and a Pi transporter inhibitor (PFA), there is preservation of ∆Ψm, glutathione levels, and caspase activity (Table 3) [32].

## 4. Endothelial Dysfunction and Cardiovascular Disease

Hyperphosphataemia has recently been recognized as an essential factor in developing medial calcification by inducing the differentiation of vascular smooth muscle cells (VSMCs) into osteoblast-like cells [35]. Studies that examined the response of aortic smooth muscle cells exposed to high Pi levels (similar to those seen in CDK patients, >1.4 mM Pi) show dose-dependent increases in cell culture calcium deposition [36]. Thus, Pi-induced changes include increased expression of the osteogenic markers osteocalcin and core-binding factor-1 genes, the latter of which is considered a “master gene” critical for osteoblast differentiation [35]. Therefore, hyperphosphataemia can be a risk factor for atherosclerosis, resulting in cardiovascular disease (CVD).

It has been reported that oxidative stress and the resulting endothelial dysfunction play a key role in the pathogenesis of atherosclerosis and CVD [37,38]. Excess mitochondrial ROS precedes and promotes atherosclerosis [39]. Some studies have shown that high Pi levels promote oxidative stress, impairing endothelial function, as one of the causes of CVD (Table 4) [40,41,42,43,44,45,46].

Some studies using bovine aortic endothelial cells (BAECs) from the descending thoracic aorta of bovine foetus to assess endothelial function showed that Pi overload results in the generation of ROS (2.8 mM treatment for 80 min) and reduces NO production (1.8 mM for 600 and 800 s). Regarding Pi-induced ROS production, the NADPH oxidase inhibitor DPI abolishes ROS production, suggesting the participation of this enzyme in this process (Table 4) [40]. The same group showed that the sodium-dependent Pi transporter precedes Pi-induced endothelial dysfunction and the activation of protein kinase C (PKC) (Table 4); however, it has not been elucidated whether it precedes the production of ROS if it occurs after the production of Pi-induced ROS [41].

The transdifferentiation of VSMCs primarily causes medial arterial calcification into osteoblast-like cells in hyperphosphataemia [42]. A study using an in vitro calcification model (β-glycerophosphate (BGP) induction) in bovine aortic smooth muscle cells (BASMCs) showed that the production of mitochondrial ROS, or superoxide anion, is stimulated by increased mitochondrial membrane potential (Table 4). Calcium deposition and the switch of smooth muscle cells from a contractile to an osteogenic phenotype are observed after BGP-induced mitochondrial ROS generation [42].

Later, it was shown that calcifying VSMCs treated with Pi (3.6 mM) after 4 days exhibit mitochondrial dysfunction by a decreased mitochondrial membrane potential. In addition, ATP production and increased ROS production are accompanied by mitochondrial-dependent apoptotic events, including (1) release of cytochrome c from the mitochondria into the cytosol; (2) caspase-9 and caspase-3 activation; and (3) chromosomal DNA fragmentation [44]. However, it was demonstrated that α-lipoic acid (1,2-dithiolane-3-pentanoic acid, ALA), a potent antioxidant soluble in both fat and water, is able to block Pi-induced VSMC apoptosis and calcification by restoring mitochondrial function and intracellular redox status (Table 4) [43].

In human endothelial cells (EAhy926 cells and GM-7373 cells), high phosphate concentrations (similar to uraemia-associated hyperphosphataemia, >2.5 mM Pi) induce cell apoptosis (Table 4) [40]. This effect is enhanced when cells are incubated for 24 h in the presence of 2.8 mM calcium instead of 1.8 mM calcium. The process of phosphate-induced apoptosis is further characterized by increased oxidative stress, as detected by increased ROS generation and disruption of the mitochondrial membrane potential at 2 h after treatment, followed by caspase activation (Table 4). This effect is dependent on the sodium-dependent Pi transporter after testing showed a reversible effect of the disruption of the mitochondrial membrane potential in the presence of PFA (Table 4) [44].

In addition, increased cytosolic Ca^2+^ and Pi-induced oxidative stress are indispensable for osteogenic differentiation and calcification in vascular smooth muscle. In primary vascular smooth muscle cells and A7r5 cells, it has been demonstrated that high extracellular Pi increases the intracellular Ca^2+^ concentration [Ca^2+^]_i_ via voltage-gated Ca^2+^ entry triggered by depolarization of plasma membrane potential (ΔΨp) [45]. The [Ca^2+^]_i_ increase by high Pi is responsible for oxidative stress and calcification in vascular smooth muscle [45].

In human umbilical vein endothelial cells (HUVECs) and C57Bl/6 mice, high extracellular phosphate impairs arterial endothelial function by PPARα/LKB1/AMPK/ NOS pathway inhibition (Table 4) [46]. Physiologically, arterial endothelial function is activated by peroxisome proliferator-activated receptor α (PPARα), which improves flow-mediated dilation and protects against endothelial dysfunction and other hypertensive effects by increasing the expression and activity of endothelial NOS [46]. This study showed that a high extracellular Pi concentration is transported into endothelial cells through PiT-1 and reduces endothelial NOS activity via the PPARα/LKB1/AMPK/NOS pathways. Inhibition of AMPK reduces PGC1-α and antioxidant gene expression and then increases ROS production. The inhibited NOS activity and increased ROS production decrease NO bioavailability, thereby leading to vascular endothelial dysfunction (Table 4) [46].

## 5. High Phosphate Induces Impaired Insulin Secretion

Pancreatic β-cells act as fuel sensors and secrete insulin to lower blood glucose levels [47,48]. In pancreatic β-cells, glucose metabolism and oxidative phosphorylation increase the ATP/ADP ratio, inhibiting K_ATP_ channel activity. The resulting depolarization of the plasma membrane initiates voltage-sensitive Ca^2+^ influx, the primary signal for insulin granule exocytosis [47,48]. During metabolism-secretion coupling induced by nutrients, hyperpolarization of the electric gradient occurs in the inner mitochondrial membrane, increasing the driving force of ATP synthase for the synthesis of ATP from ADP and inorganic phosphate [47]. In addition, this hyperpolarization promotes and accelerates the export of ATP from the matrix space to the cytosol catalysed by adenosine nucleotide (ANT; ATP_4_^−^ transport is exchanged for ADP_3_^−^) [47,48,49,50].

In addition to the electrical gradient, the chemical gradient of H^+^ (the pH in the mitochondrial matrix is higher than that in the cytosol) increases the driving force of ATP synthase and boosts transport coupled to protons of metabolites or ions, such as pyruvate or Pi [49]. Regarding phosphate, cytosolic Pi enters the mitochondrial matrix together with H^+^ or in exchange for OH at the expense of the proton gradient [51]. In rat insulinoma cells (INS-1E) permeabilized by a-toxin, succinate and glycerol-3-phosphate, dependent on exogenous ADP and Pi, cause acidification and hyperpolarization of the mitochondrial matrix, which promotes the export of ATP [52]. In addition, the dissipation of the mitochondrial pH gradient or the pharmacological inhibition of Pi transport blocks the effects of Pi on the electrochemical gradient and the export of ATP [52].

It has been shown that individuals with a high plasma Pi level are at an increased risk of developing type 2 diabetes (Figure 3) [53]. Some studies have demonstrated the harmful effect of Pi on insulin secretion in chronic renal patients associated with hyperphosphataemia and “Klotho”-deficient mice (kl/kl) [54,55]. Due to previous reports that high extracellular Pi induces ROS production, it has been shown that in INS-1E cells with high extracellular Pi (5 mM treatment for 24 h), there is an increase in mitochondrial membrane potential, superoxide generation, caspase activation, and cell death (Figure 3, Table 4).

The participation of the mitochondrial Pi transporter is associated with the observation of cell death, hyperpolarization, and Pi-induced ROS production being prevented by mitochondrial transporter (BMA—butylmalonate) inhibitors and mitochondrial antioxidants mitoTEMPO or MnTBAP (Table 5) [56]. Additionally, it has been demonstrated that high Pi exposure upregulates PiT-1 and PiT-2 expression and causes cytosolic alkalinization. This increase in intracellular pH facilitates Pi transport into the mitochondrial matrix and subsequently accelerates superoxide production, mitochondrial permeability transition, mithocondrial dysfunction, and defective insulin secretion [57].

It has been suggested that ROS elicits endoplasmic reticulum (ER) stress, including the unfolded proteins response (UPR), which exacerbate type 2 diabetes. For this reason, the correlation of Pi-induced oxidative stress with RE stress has been demonstrated with impaired insulin release [57,62,63]. In rat insulinoma cells (INS-1E), the mitochondrial oxidative stress induced by high Pi showed a perturbed endoplasmic reticulum by UPR activation (PERK and eIF2a). In addition, the ER stress induced by UPR decreases insulin content (Figure 3) [57].

Therefore Pi-induced ROS production initiates ER stress, causing UPR resulting in insulin translation attenuation, which lowers insulin content [57]. In addition, elevated extracellular Pi decreases ATP content, cytosolic Ca^2+^ oscillations, insulin content, and secretion in INS-1E cells. However, these effects were restored after preincubation with mitochondrial antioxidants, suggesting that elevated extracellular Pi causes mitochondrial oxidative stress linked to mitochondrial hyperpolarization. This stress results in reduced insulin content and secretion in hyperphosphataemic states [56,57].

## 6. Skeletal Muscle Atrophy and Suppressed Myogenic Differentiation

The loss of skeletal muscle represents a typical ageing phenotype and pathological conditions, such as CKD [58]. Skeletal muscle can receive internal and external stimuli such as hormones, cytokines, and nutrients, shifting the balance between protein synthesis and degradation and determining muscle fibre size and contractile function. An imbalance in proteostasis is observed in several catabolic conditions, such as sarcopenia, cachexia, and disuse, leading to muscle atrophy [58].

It has recently been shown that exposing C2C12 skeletal muscle cells to high Pi leads to changes consistent with muscle wasting, mainly regulated by ROS production. The authors pointed to ROS generation by a mitochondrial source after reducing the mitochondrial membrane potential (Table 5) [61].

Pi-induced ROS production promotes a reduced myotube size, decreases protein synthesis markers (mTOR and S6K), and enhances protein degradation markers (MuRF1 and atrogin-1). In addition, it was shown that high Pi suppresses myogenic differentiation by reducing myogenic differentiation markers (myogenin, MYH, and troponin I) and promotes muscle atrophy through oxidative stress-mediated protein degradation and both canonical (ROS-mediated) and noncanonical (p62-mediated) activation of Nrf2 signalling [58].

## 7. High Pi Diet and Male Reproductive System Dysfunction

Several reports have demonstrated high serum phosphate levels in cardiac calcification and bone diseases, as previously described in this work. In addition, the level of phosphorus has been found to be positively correlated with a predictive factor for erectile dysfunction and the inability to obtain or maintain an erection [64].

Banjoko et al. [65] analysed the pH of seminal human plasma and observed a low Pi concentration in sperm groups with low motility compared to the regular motility group [65]. The influence of Pi concentration on the function of the male reproductive system was further investigated using Wistar Kyoto rats with a regular diet (0.3% phosphorus) compared to a high-phosphorus diet (1.2% phosphorus). With the increase in oxidative stress and apoptosis in the high-phosphorus group, the effect of overload is relatively obvious. The study also analysed a group of mice fed high phosphorus that drank a potent ROS scavenger, N-acetylcysteine (NAC), at 1.5 mg/weight/day (Figure 4 and Table 5) [60].

A high Pi diet in mice promotes a decrease in testicular weight, sperm count, sperm motilities, and seminiferous tubule diameter, but this effect is restored by supplementation with NAC. In addition, the apoptotic rate is high for mice fed a high Pi diet. Nevertheless, NAC is able to block this effect, suggesting that reactive Pi-induced species participate in this effect in male reproductive system dysfunction in mice. This suggestion was better supported after observing that the level of MDA (malondialdehyde, a classic marker of oxidative stress for male infertility studies) is higher in a diet with high Pi but restored in the presence of NAC in the testicular tissue of mice (Table 5) [60].

Another study sought to relate the influence of high Pi on male reproductive function in C57BL/6 mice with CKD induced after nephrectomy [59]. Therefore, the mice were randomly divided into four groups: (1) normal food (0.6% phosphorus); (2) food with high Pi (2% phosphorus); (3) CKD induced with regular feeding; and (4) CKD induced with high Pi feeding. In this study, impaired testicular function and spermatogenesis were observed, possibly associated with oxidative stress observed by the high production of MDA and a low level of antioxidant activities (SOD, CAT, and GPx) in mice with induced CKD. In addition, the high Pi diet aggravates the negative effects of testicular damage in CKD-induced mice (Figure 4 and Table 5) [59]. Therefore, all of these results suggest that supplementation with phosphate may aggravate damage to male reproductive system function, either in CKD patients or in healthy individuals with risk factors.

## 8. Conclusions

In the human body, most phosphate is present in adjacent bones and tissues, with the remainder distributed among other tissues and extracellular fluid. In bone tissues and cartilage, a high concentration of Pi acts on osteoblasts and chondrocytes as an apoptotic signal after triggering the production of ROS (Figure 5). In addition, Pi-induced ROS production can induce endothelial dysfunction, leading to vascular calcification as a risk factor for cardiovascular diseases (Figure 5). In other tissues, β-pancreatic cells have impaired insulin release when ROS production is induced by a high concentration of Pi, strengthening the hypothesis that individuals with hyperphosphataemia have an increased risk of developing diabetes (Figure 5). In addition, hyperphosphataemia caused by CKD has been strongly associated with muscle atrophy and impaired male reproductive function through oxidative stress induced by high Pi (Figure 5). Interestingly, most studies regarding induced ROS production show that the failure of Pi-induced biological function can be reversed in the presence of different antioxidants or inhibitors of cytoplasmic and mitochondrial Pi transporters, showing the pharmacological prospects for regulating these pathological conditions when extracellular Pi increases ROS.

## Figures and Tables

**Figure 1 ijms-22-07768-f001:**
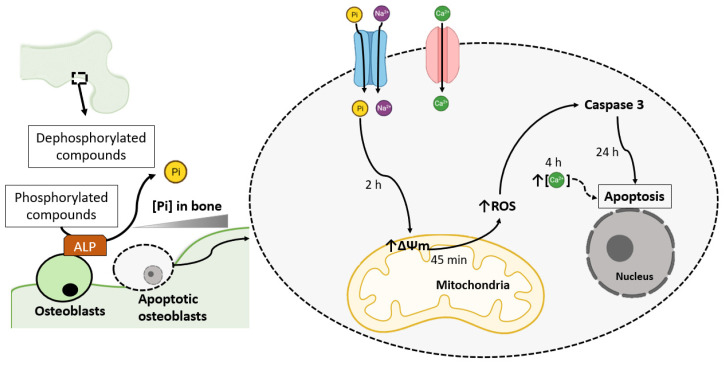
Phosphate-induced osteoblast apoptosis mediated by ROS production. In the localized remodelling area, when osteoblasts are exposed to Pi levels substantially more significant than those typically found in serum, osteoblasts initiate apoptotic pathways by uncontrolled ALP activity. Sodium-dependent Pi transporters internalize Pi. Pi promotes a hyperpolarization of the mitochondrial membrane potential (∆Ψm) until 2 h and induces ROS production. ROS can activate caspases. Ca^2+^ can be internalized by Ca^2+^ channels (although this calcium channel has little effect on Pi-induced osteoblast apoptosis signalling). High intracellular Ca^2+^ is observed after 4 h by Pi treatment, and Ca^2+^ may influence Pi-mediated bone cell apoptosis [19,22,23,24]. Solid black arrows indicate activation, and dashed black arrows indicate possible activation.

**Figure 2 ijms-22-07768-f002:**
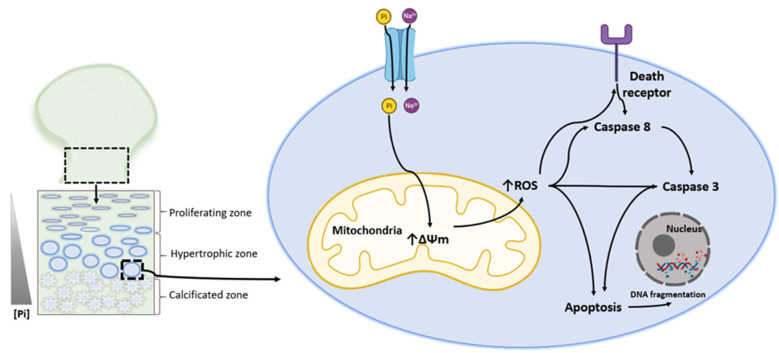
Phosphate-induced chondrocyte apoptosis is mediated by ROS production. In the growth bone plate, the cells terminally differentiated before zone calcification undergo apoptosis or programmed cell death. An increase in Pi levels is noted [25,26]. Sodium-dependent Pi transporters internalize Pi. Pi promotes a hyperpolarization of the mitochondrial membrane, which induces the production of ROS. ROS can activate caspases or death receptors and thus promote apoptosis, including DNA fragmentation [28,29,30,31]. Solid black arrows indicate activation.

**Figure 3 ijms-22-07768-f003:**
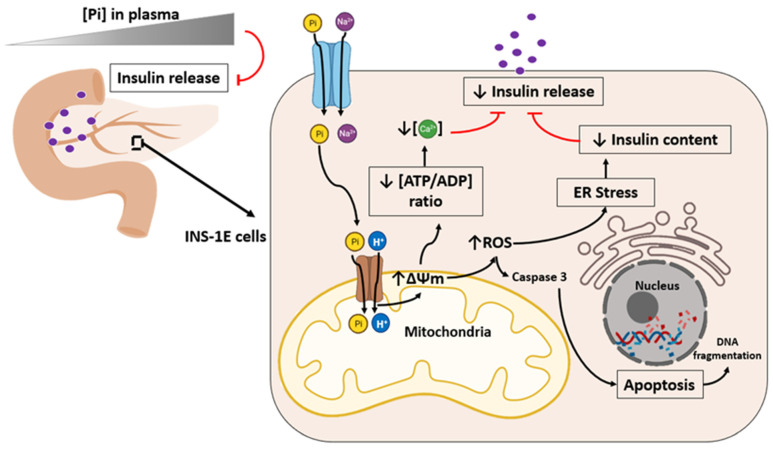
High phosphate induces ROS production and impairs insulin secretion. It has been demonstrated that decreased insulin content and secretion are correlated under hyperphosphataemic states (elevated plasma Pi concentrations). Elevated extracellular Pi (1–5 mM) increases the mitochondrial membrane potential (∆Ψm), superoxide generation, caspase activation, and cell death [56]. Elevated extracellular Pi diminishes ATP synthesis, cytosolic Ca^2+^ oscillations, and insulin content and secretion in INS-1E cells and dispersed islet cells [56]. In addition, Pi-induced superoxide production promotes endoplasmic reticulum (ER) stress, which decreases insulin content, impairing insulin release [57]. Solid black arrows indicate activation. Solid red traces indicate inhibition.

**Figure 4 ijms-22-07768-f004:**
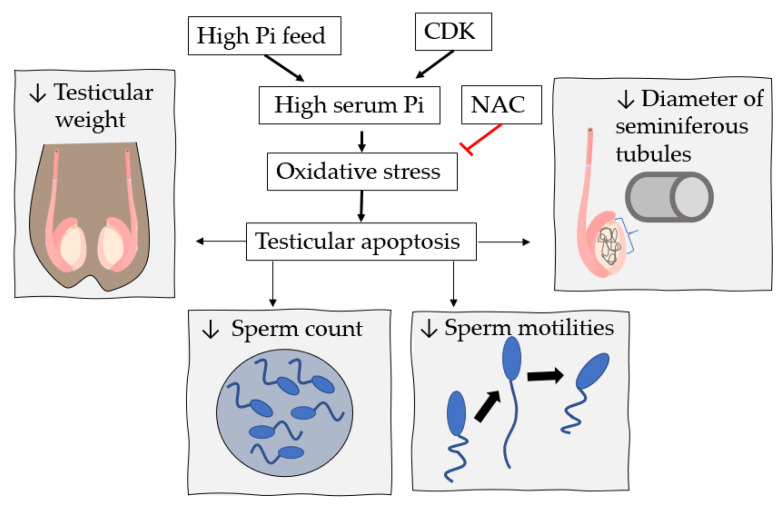
High Pi impairs testicular function by oxidative stress. A high Pi diet, chronic kidney disease (CKD), or both in WKY rats [60] or C57BL/6 mice [59] increases serum Pi, oxidative stress, and apoptosis in the testicles. Testicular apoptosis induced by oxidative stress promotes a decrease in testicular weight, sperm count, sperm motilities, and seminiferous tubule diameter [59,60]. Nevertheless, this effect is restored by supplementation with a potent oxidative species scavenger, N-acetylcysteine (NAC) [59].

**Figure 5 ijms-22-07768-f005:**
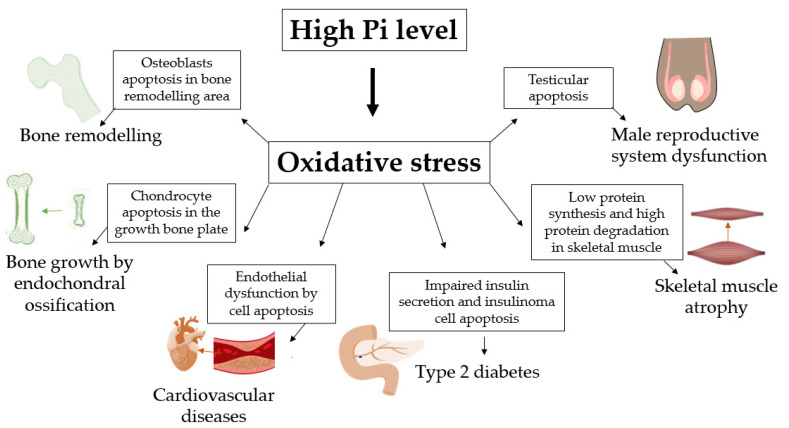
Oxidative stress high Pi-induced regulates physiological processes and disease development. High Pi level can induce oxidative stress and regulates bone remodeling [22,23,24], bone growth by endochondral ossification [28,29,30,31], cardiovascular diseases by atherosclerosis [40,41,42,43,44,45,46], type 2 diabetes [56,57], skeletal muscle atrophy [61] and male reproductive system dysfunction [59,60].

**Table 1 ijms-22-07768-t001:** Foods with high and low estimated total dietary phosphorus (mg/day) [5]. The food group was classified according to the United States Department of Agriculture (USDA).

High Phosphorus ^1^	Low Phosphorus ^2^
Grain Products.	Sugar, Sweeteners & Beverages.
Meat, Poultry, Fish & Mixtures.	Vegetables and fruits.
Milk & Milk Products.	Legumes, Nuts, Seeds and eggs.
	Fats, Oils & Salad Dressings.

^1^ Foods with high phosphorus (>20 mg/day); ^2^ Foods with low phosphorus (>7.1 mg/day).

**Table 2 ijms-22-07768-t002:** Pi-induced ROS production in osteoblasts cells.

Cell Type or Tissue Sample	Results Observed	Pi Treatment	Pharmacological Effect
Osteoblastic murine MC3T3-E1 cells	Increased ROS production by NOX 1 and 4 [20].	5 mM Pi for 20–42 h [20].	Inhibition by NOX inhibitors (0.5 mM apocynin and 10 μM DPI) and PFA (0.5–1 mM) [20].
	Low osteoblastic markers expression (ALP, osteocalcin, and runt-related transcription factor 2) [20].	5 mM Pi for 48 h [20].	-
Rat osteoblastic cell line (UMR-106).	Increased ROS production [21].	5 mM Pi for 48 h [21].	Inhibition by 0.5 mM apocynin [21].
	Increased FGF-23 release [21].	5 mM Pi for 48 h [21].	Inhibition by 0.5 mM apocynin [21].
Osteoblast-like cells (human bone)	Increased mitochondrial membrane potential [22,23].	7 mM Pi for 24 h [22]. 5 mM Pi and 2.9 mM Ca^2+^ for 2 h [23].	-
	Increased ROS production [22].	8 mM Pi and 2.9 mM Ca^2+^ for 45 min [24].	-
	Increased intracellular Ca^2+^ [23].	5 mM Pi and 2.9 mM Ca^2+^ for 4 h [23].	-
	Increased cell death [22,23].	7 mM Pi for 24, 48 and 96 h [22]. 5 mM Pi and 2.9 mM Ca^2+^ for 24 h [23].	Inhibition by Pi transporter inhibitor (PFA, 5 mM) [22,23] and apoptosis inhibitors (aurintricarboxylic acid, ATA or DEVD-CHO) [21].
	Increased cell apoptosis [20,21].	7 mM Pi for 48 h [20]. 3 mM Pi and 2.9 mM Ca^2+^ for 24 h [21].	-

**Table 3 ijms-22-07768-t003:** Effects of high extracellular Pi in chondrocytes from the proximal heads of chick embryo tibias.

Results Observed	Pi Treatment	Pharmacologic Effect
A marked early increase in mitochondrial membrane potential until 135 min and a small decrease after 150 min [31].	3 mM Pi and 2.8 mM Ca^2+^ [31].	-
Decreased mitochondrial membrane potential [29,32].	7 mM Pi overnight [28].	-
	5 mM Pi for 24 h [32].	Inhibition by Pi transporter inhibitor (5 mM PFA) and 5 mM NOS inhibitors (L-NAME and L-NMMA) [32].
Increased ROS production after 75 min [31].	3 mM Pi and 2.8 mM Ca^2+^ [31].	-
Increased intracellular Ca^2+^ after 45 min [31].	3 mM Pi and 2.8 mM Ca^2+^ [31].	-
Increased nitric oxide generation [32].	5 mM Pi for 24 h [32].	Inhibition by 5 mM PFA and L-NAME and L-NMMA [32].
Decreased glutathione levels [32].	5 mM Pi for 24 h [32].	Inhibition by 5 mM PFA and 5 mM L-NAME [32].
Increased cell apoptosis [28,29,32].	7 mM Pi overnight [28,29].	Inhibition by 1 mM PFA and alendronate [28,29].
	5 mM Pi for 24 h [32].	Inhibition by 5 mM PFA and 5 mM L-NAME and L-NMMA [32].

**Table 4 ijms-22-07768-t004:** Pi-induced ROS production is a possible risk factor for cardiovascular disease.

Cell Type or Tissue Sample	Results Observed	Pi Treatment	Pharmacologic Effect
Bovine aortic endothelial cells [40,41].	Increased ROS production [40].	2.8 mM Pi for 80 min.	Inhibition by NOX inhibitor (10 μM DPI).
	Reduced NO production [40].	2.8 mM Pi for 800 s.	-
	Increased PKC activity [41].	3 mM Pi for 15 min.	Inhibition by 200 μM PFA.
Bovine aortic smooth muscle cells [42].	Increased mitochondrial membrane potential.	10 mM β-glycerophosphate (BGP) for 2 days.	-
	Increased ROS production	10 mM BGP for 2 days.	Inhibition by a respiratory chain inhibitor (rotenone, 10 μM) or carbonyl cyanide m-chlorophenyl hydrazone (CCCP; 10 μM).
	Increased smooth muscle cells calcification and bone-related markers.	10 mM BGP for 2 days.	Inhibition by 10 μM rotenone and 10 μM CCCP.
Bovine aortic smooth muscle cells from male C57BL6N mice [43].	Decreased mitochondrial membrane potential.	3.6 mM Pi for 4 days.	-
	Decreased ATP production.	3.6 mM Pi for 4 days.	Inhibition by 300 μM α-lipoic acid.
	Increased ROS production.	3.6 mM Pi for 4 days.	Inhibition by 300 μM α-lipoic acid.
	Increased cells apoptosis.	3.6 mM Pi for 4 days.	Inhibition by 300 μM α-lipoic acid.
	Increased smooth muscle calcification.	3.6 mM Pi for 4 days.	Inhibition by 300 μM α-lipoic acid.
Human endothelial cells (EAhy926 cells and GM-7373 cells) [44].	Decreased mitochondrial membrane potential.	2.5–5 mM Pi and 2.8 mM Ca^2+^ for 2 h.	Inhibition by a Pi transporter inhibitor (1 mM PFA).
	Increased ROS production.	2.5 mM Pi and 2.8 mM Ca^2+^ for 75 min.	Inhibition by a superoxide scavenger (3-dimethyl-2-thiourea—DMTU, 10 mM).
	Increased cell apoptosis.	5 mM Pi and 2.8 mM Ca^2+^ for 24 h.	Inhibition by a general caspase inhibitor (Z-VAD-FMK, 100 μM).
Human umbilical vein endothelial cells (HUVECs) and endothelial cells of C57Bl/6 mice [46].	Impairment of the PPARα/LKB1/AMPK/NOS pathway.	3 mM Pi for 48 h.	Inhibition by a AMPK agonist (AIACR, 100 μM), PPARα agonist (100 μM WY-14643), PGC-1α inhibitor (SR-18292, 50 µM) and PFA (0.5 mM).
	Decreased the mitochondrial membrane potential.	3 mM Pi for 48 h.	Inhibition by AIACR (100 μM), SR-18292 (50 µM).
	Increased mitochondrial ROS production.	3 mM Pi for 48 h.	Inhibition by AIACR (100 μM), SR-18292 (50 µM), cytoplasmic ROS scavenger (tempol, 100 µM) and mitochondrial ROS scavenger (mito-tempo, 100 µM).
	Reduced NO production.	3 mM Pi for 48 h.	Inhibition by AIACR (100 μM), WY-14643 (100 μM).
	Increased relaxation in mesenteric arteries.	Mice fed high Pi diet (1.3% phosphate).	Inhibition by AIACR (100 μmol/L), compound C (1 μmol/L, AMPK inhibitor) and WY-14643 (100 μmol/L in drinking water, 2 weeks).

**Table 5 ijms-22-07768-t005:** Effects of Pi-induced ROS production in different tissues.

Cell Type or Tissue Sample	Results Observed	Pi Treatment	Pharmacologic Effect
Rat insulinoma cells [56].	Increased the mitochondrial membrane potential.	3–5 mM Pi for 30 min.	Inhibition by FCCP ionophore, butylmalonate (BMA).
	Increased superoxide generation.	3–5 mM Pi for 30 min.	Inhibition by BMA and mitochondrial antioxidants: mitoTEMPO (100 nM) or Mn-TBAP (0.5 mM).
C2C12 skeletal muscle cells [58].	Decreased the mitochondrial membrane potential.	4 mM Pi for 24 h.	-
	Increased O_2_ consumption.	4 mM Pi for 24 h.	-
	Increased ROS generation.	4 mM Pi for 24 h.	Strongly inhibited by cytosolic ROS scavenger (10 mM NAC).
	High MDA production [59,60].	Normal diet (0.3% Pi) compared to high Pi diet (1.2%) [60]. Normal diet (0.6% Pi) compared to high Pi diet (2.0%) [59].	Inhibition by NAC (1.5 mg/weight/day) [60].
Testicular tissue of WKY rats [60] or C57BL/6 mice [59].	Decreased glutathione levels [61].	High Pi diet (1.2%) [60].	-
	Low level of antioxidant activities (SOD, CAT and GPx) [60].	High Pi diet (2.0%) [59].	-

## Data Availability

The data that support the findings of this study are available from the corresponding author upon reasonable request.

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
