# Peer review of "Extracellular Inorganic Phosphate-Induced Release of Reactive Oxygen Species: Roles in Physiological Processes and Disease Development"

_ijms, 2021, doi:10.3390/ijms22157768_

Round 1
Reviewer 1 Report
Summary
This is a short review and well written review of Phosphate metabolism and the role of ROS in physiology and disease. The authors have explained the significance of diet, low, moderate and severe levels phosphate and listed the consequences in sickness and in health.
LIMITATION & STRENGTHS
The main strength is the explanation on the amounts of inorganic phosphate and regulation of mitochondrial dysfunction. The major limitation is the incomplete report of how ROS impacts endoplasmic reticulum in physiology and disease.
Suggestions for Authors
#1 Line 339
Although, the review is quite balance and succinct, the authors failed to cite more recent literature on the topic of over-nutrition leading to accumulation of UPR genes in β-cells, the effect of (ROS) signaling to insulin secretion, associated ER stress leading to diabetes. They briefly mention it in passing in line 339. The readers would benefit with expansion on this topic.
#2 Line 161-163
Yes, I agree with the explanation on the role of FGF23. Given the vast amount of literature and significance of FGF23, it would benefit the readers if the authors expanded to explain the role of “alpha klotho” in regulating phosphate levels in CKD.
#3 Line 405
It is important to detail the different foods with normal and high phosphorous content (a table or a figure) in humans as the section talks about high Pi diet male reproductive dysfunction. This will guide the readers in the right direction.
Author Response
Reviewer 1:
Question 1: Line 339 Although, the review is quite balance and succinct, the authors failed to cite more recent literature on the topic of over-nutrition leading to accumulation of UPR genes in β-cells, the effect of (ROS) signaling to insulin secretion, associated ER stress leading to diabetes. They briefly mention it in passing in line 339. The readers would benefit with expansion on this topic.
Answer: As suggested, this topic was better discussed (Page 10, lines 362-368; page 11, lines 369-370) and new references ([58] Back et al., 2012; [59] Burgos-Morón et al., 2019) were included in the revised manuscript (References, page 17, lines 620-624). According, changes were made in a new figure 3 (Page 11, lines 383-385).
Question 2: Line 161-163 Yes, I agree with the explanation on the role of FGF23. Given the vast amount of literature and significance of FGF23, it would benefit the readers if the authors expanded to explain the role of “alpha klotho” in regulating phosphate levels in CKD.
Answer: As required, the role of “alpha klotho” in regulating phosphate levels was better explained in the revised manuscript (Page 2, lines 52-53; 65-67).
Question 3: Line 405 It is important to detail the different foods with normal and high phosphorous content (a table or a figure) in humans as the section talks about high Pi diet male reproductive dysfunction. This will guide the readers in the right direction.
Answer: As required, we included new Table 1 detailing the different foods with normal and high phosphorus in the revised manuscript (Page 2, lines 30-31, Page 3, lines 91-95). According, we included new reference ([5] McClure et al., 2017, Page 15, lines 497-498).
Reviewer 2 Report
Dear editors: It is a great honor and pleasure for me to be invited as the reviewer for this important work. Lacerda-Abreu et al. comprehensively reviewed the roles of extracellular inorganic phosphate (Pi)-induced Reactive Oxygen Species (ROS) release from physiological processes to disease development.
This study topic is interesting, attributing to Prof. Meyer-Fernandes’s long-term efforts and contributions in this scientific field. Although the article is well-written,
I have a number of comments concerning this study:
- The main findings and strengths of the study: the investigators demonstrated the effect of extracellular Pi-induced ROS on diverse aspects, including the promotion of osteoblast apoptosis, chondrocyte apoptosis, endothelial dysfunction in cardiovascular diseases, impaired insulin secretion with an increased risk of developing diabetes, skeletal muscle atrophy along with suppressed myogenic differentiation, and high Pi diet related male reproductive system dysfunction.
- Line 56-58: Hyperphosphataemia is one of the main causes of morbidity and mortality in patients with CKD and can also be a cause of acute kidney injury (AKI) [12]. New citation was required because the “Reference 12” did not report that hyperphosphataemia can also be a cause of AKI. On the contrary, hyperphosphataemia often refers to the result of AKI. In rare conditions, hyperphosphataemia could cause AKI. Thus the word “in certain scenarios” should be added at the end of the sentence.
- Line 70: The sentence “two reactive oxygen species” seems redundant that should be deleted.
- Line 76: Concerning the use of abbreviation of “ROS” for the first time in Line 69-70, the term of “reactive oxygen species” should be should be replaced by “ROS” here and in Line 418 and 429.
- In light of the similar problems as above, please check other abbreviation uses, eg., “cardiovascular disease (CVD)” in Line 245 and 247; “chronic kidney disease” in Line 357, 404 and 423. Please add the abbreviation list after the body text, and all the important abbreviations in the text should be presented at the end of the text.
- The resolution of figures seems suboptimal, and extensive editing of English language and style are required Thank you for giving me the opportunity to review this interesting article. After appropriate revision, this important review article should be published as soon as possible.
Author Response
Reviewer 2:
Question 1: Line 56-58: Hyperphosphataemia is one of the main causes of morbidity and mortality in patients with CKD and can also be a cause of acute kidney injury (AKI) [12]. New citation was required because the “Reference 12” did not report that hyperphosphataemia can also be a cause of AKI. On the contrary, hyperphosphataemia often refers to the result of AKI. In rare conditions, hyperphosphataemia could cause AKI. Thus the word “in certain scenarios” should be added at the end of the sentence.
Answer: We agreed, and deleted the mention regarding acute kidney injury (AKI) in the revised manuscript (Page 2, lines 62-63).
Question 2: Line 70: The sentence “two reactive oxygen species” seems redundant that should be deleted.
Answer: As required, we deleted the sentence “two reactive oxygen species” in the revised manuscript (Page 2, line 77).
Question 3: Line 76: Concerning the use of abbreviation of “ROS” for the first time in Line 69-70, the term of “reactive oxygen species” should be should be replaced by “ROS” here and in Line 418 and 429.
Answer: As suggested, we replaced the term of “reactive oxygen species” by “ROS” in the revised manuscript (Page 3, line 83; page 13, line 452; page 14, line 464).
Question 4: In light of the similar problems as above, please check other abbreviation uses, eg., “cardiovascular disease (CVD)” in Line 245 and 247; “chronic kidney disease” in Line 357, 404 and 423. Please add the abbreviation list after the body text, and all the important abbreviations in the text should be presented at the end of the text.
Answer: As required, others abbreviation was checked in the revised manuscript (Page 8, lines 268, 271; page 12, line 390; page 13, line 437; page 14, line 458).
Question 5: The resolution of figures seems suboptimal, and extensive editing of English language and style are required.
Answer: As suggested, the resolution of figures was improved. Regarding English language and style, the text had already passed by the American Journal Experts (AJE) correction before this submission (certificate attached). As required, we modified the text and improved English writing in revised manuscript.

Reviewer 3 Report
This is a potentially interesting manuscript that focuses on the connection of inorganic phosphate, ROS and cell death and the implication of this axis in diseases. Overall, the manuscript would benefit from less detailed experimental descriptions and more focus on the higher-level understanding of molecular pathways. This could be aided by improving the organization of the manuscript within the sections.
1.General organization of the manuscript: For each section, a short description of the most common model systems would be beneficial for the reader. For example, Section 3 (3. Phosphate-induced chondrocyte apoptosis is mediated by ROS production) could start with the general description of the chondrocytes and their importance in bone differentiation, as it is in the present form of the manuscript. It could then be followed by the description of the most common experimental model system, the proximal heads of chick embryo tibias. Description of individual experiments would follow.
I also suggest to group the overview of experiments according to their general theme and to discuss their relevance for human health at the end of the section. For example, Section 4. (4. Endothelial Dysfunction and Cardiovascular Disease) seems to have a particularly high clinical relevance. Would it make sense to discuss the bovine/mouse models first and after proceed to human diseases?
I also suggest that the authors prepare a table or figure that focuses on the clinical relevance (associated diseases), as a ‘take home message’.
- Several times I find the description of the experiments too detailed, including unnecessary technical details. The review has lots of information and I think less details and better organization would be beneficial. Groups of experiments or themes should be discussed to provide the reader with a higher-level overview and a better overall understanding.
- Sometimes the experimental description or their conclusion seem unclear. For example Lines 192-197: The logical steps to arrive at the conclusion of the described experiment are unclear (Ref 28).
Author Response
Reviewer 3:
Question 1: General organization of the manuscript: For each section, a short description of the most common model systems would be beneficial for the reader. For example, Section 3 (3. Phosphate-induced chondrocyte apoptosis is mediated by ROS production) could start with the general description of the chondrocytes and their importance in bone differentiation, as it is in the present form of the manuscript. It could then be followed by the description of the most common experimental model system, the proximal heads of chick embryo tibias. Description of individual experiments would follow. I also suggest to group the overview of experiments according to their general theme and to discuss their relevance for human health at the end of the section. For example, Section 4. (4. Endothelial Dysfunction and Cardiovascular Disease) seems to have a particularly high clinical relevance. Would it make sense to discuss the bovine/mouse models first and after proceed to human diseases? I also suggest that the authors prepare a table or figure that focuses on the clinical relevance (associated diseases), as a ‘take home message’.
Answer: As required, the short description of the most common model systems was added to the revised manuscript (page 6, lines 199-201; page 8, lines 272-273). In addition, the overview of the experiments was grouped according to their general theme and model used in the revised manuscript (Page 3, lines 110-114, page 4, lines 118-135; page 10, lines 362-368; page 11, lines 369-370; changes in references order, page 16, lines 527-532, 572-586, lines 616-624). As suggested, new figure 5 was included in the revised manuscript (Page 14, lines 465-469) focused in physiological processes and disease development.
Question 2: Several times I find the description of the experiments too detailed, including unnecessary technical details. The review has lots of information and I think less details and better organization would be beneficial. Groups of experiments or themes should be discussed to provide the reader with a higher-level overview and a better overall understanding.
Answer: As suggested, we deleted the informations regarding technical details and experiments in the revised manuscript, to provide a better overall understanding (Page 4, lines 141-142; Page 5, lines 154-157, 167-170; Page 6, lines 212, 220-222; page 13, line 429).
Question 3: Sometimes the experimental description or their conclusion seem unclear. For example Lines 192-197: The logical steps to arrive at the conclusion of the described experiment are unclear (Ref 28).
Answer: The non-experimental data (represented by dashed lines) were removed in Figure 2 (Page 6, lines 208-211). According, the new Figure 2 was included in the revised manuscript with experimental data only.
Round 2
Reviewer 1 Report
The author's have adequately responded to reviewer's comments. The manuscript is now acceptable for publication.